# Compositional 3D-aware Video Generation with LLM Director

**Hanxin Zhu**[1]*, **Tianyu He**[2], **Anni Tang**[3], **Junliang Guo**[2], **Zhibo Chen**[1], **Jiang Bian**[2]

[1]University of Science and Technology of China
[2]Microsoft Research Asia
[3]Shanghai Jiao Tong University

hanxinzhu@mail.ustc.edu.cn, tianyuhe@microsoft.com, memory97@sjtu.edu.cn,
junliangguo@microsoft.com, chenzhibo@ustc.edu.cn, jiang.bian@microsoft.com

## Abstract

Significant progress has been made in text-to-video generation through the use of powerful generative models and large-scale internet data. However, substantial challenges remain in precisely controlling individual concepts within the generated video, such as the motion and appearance of specific characters and the movement of viewpoints. In this work, we propose a novel paradigm that generates each concept in 3D representation separately and then composes them with priors from Large Language Models (LLM) and 2D diffusion models. Specifically, given an input textual prompt, our scheme consists of three stages: 1) We leverage LLM as the director to first decompose the complex query into several sub-prompts that indicate individual concepts within the video (*e.g.*, scene, objects, motions), then we let LLM to invoke pre-trained expert models to obtain corresponding 3D representations of concepts. 2) To compose these representations, we prompt multi-modal LLM to produce coarse guidance on the scales and coordinates of trajectories for the objects. 3) To make the generated frames adhere to natural image distribution, we further leverage 2D diffusion priors and use Score Distillation Sampling to refine the composition. Extensive experiments demonstrate that our method can generate high-fidelity videos from text with diverse motion and flexible control over each concept. Project page: https://aka.ms/c3v.

## 1 Introduction

Benefitting from large-scale data and the advancement of the generative models [1, 2], we have witnessed plenty of astonishing results across a wide array of tasks. For example, Large Language Models (LLM) pre-trained on web-scale datasets are revolutionizing machine learning with strong capability of zero-shot learning [3] and planning [4, 5], while diffusion models [6] empower text-to-image generation with a rapid surge in both quality and diversity [7–9].

To harness the power of text-to-image models in text-to-video generation, modern solutions directly view video as multiple images. In this way, tremendous efforts have been dedicated to extending text-to-image models with temporal interaction to ensure consistency between frames [10–17]. However, generating visual content conditioned on the textual prompt alone struggles to express multiple concepts with precise spatial layout control [18–20]. To tackle this issue, LVD [21] and VideoDirectorGPT [22] propose to first generate spatiotemporal bounding boxes of each object based on the textual prompt with LLM, and then condition the video generation on the obtained layouts. Although rough layout control can be realized, they still have inherent limitations for detailed concept control, *e.g.*, the motion and appearance of specific characters, and the movement of viewpoints.

---

*This work is accomplished in Microsoft, April 2024.

In nature, our understanding of the world is compositional [23, 24, 20], and the interaction with the world takes place in a 3D. Motivated by this, in contrast to the prior endeavors that *implicitly* learn different concepts in 2D space, we are interested in exploring an alternative solution that *explicitly* composes concepts in 3D space for video generation. To this end, we in particular identify two key technical challenges: 1) Since a textual prompt contains multiple concepts, how to coordinate the generation of various concepts? 2) Given the generated concepts, how to compose them to follow common sense in the real world?

In this work, we introduce text-guided compositional 3D-aware video generation (C3V), a novel paradigm that regards LLM as director and 3D as structural representation for video generation. C3V consists of three main stages: **1)** Given a textual prompt, to coordinate the generation of various concepts, we leverage LLM to disassemble the input prompt into sub-prompts, where each sub-prompt describes an individual concept, *e.g.*, the scene, objects, and motion. For each concept, a pre-trained expert model is assigned by LLM to generate its corresponding 3D representation (*e.g.*, 3D Gaussians [25], SMPL parameters [26]) according to the textual description. **2)** To provide coarse instruction for composition (*i.e.*, the scale and trajectory of each object in the scene), we further resort to the priors in multi-modal LLM by querying it with the rendered scene image and the textual goals. However, directly instructing multi-modal LLM to return the scale and trajectory of each object leads to unexpected results, as it is challenging for LLM to estimate visual dynamics. Therefore, we follow a step-by-step reasoning philosophy [27] by representing the object with the bounding boxes and dividing the trajectory estimation into sub-tasks, *i.e.*, estimating the starting points, ending points, and trajectories step-by-step. **3)** After obtaining the coarse trajectories from the language space, we also propose to refine the scales, rotations, and exact locations with priors from large-scale visual data. Specifically, taking inspiration from DreamFusion [28], which proposes to distill generative priors from pre-trained image diffusion models into 3D objects, we employ Score Distillation Sampling (SDS) [28] to optimize the transformation matrix of each object in 3D space.

Our system has three main advantages: 1) Because each concept is represented by individual 3D representations, it naturally supports flexible control and interaction of each concept. 2) It inherently excels at synthesizing complex and long videos such as drama, etc. 3) The viewpoint is controllable.

Extensive experiments demonstrate that our proposed method can generate 3D-aware videos with diverse motion and high visual quality, even from complex queries that contain multiple concepts and relationships. We also illustrate the flexibility of C3V by editing various concepts of the generated videos. The generated videos are presented on our project page. To the best of our knowledge, we make the first attempt towards text-guided compositional 3D-aware video generation. We hope it can inspire further explorations on the interplay between video and 3D generation.

## 2 Related Works

### 2.1 Video Generation with LLM

Recently, there have been substantial efforts in training text-to-video models on large-scale datasets with autoregressive Transformer [29, 30, 17] or diffusion models [10–13, 16]. A prominent approach for text-to-video generation is to extend a pre-trained text-to-image model by inserting temporal layers into its architecture, and fine-tuning models on video data. However, although effective, it remains challenging to generate objects with specific attributes or positions. To address this challenge, a series of studies proposed to exploit knowledge from LLM [31, 32] to achieve controllable generation [21, 19, 22, 33–35], zero-shot generation [36–39], or long video generation [40]. For example, Free-Bloom [36] and DirecT2V [38] used LLM to transform the input textual prompt into a sequence of sub-prompts that describe each frame. LVD [21] and VideoDirectorGPT [22] employed LLM to generate spatiotemporal bounding boxes to control the object-level dynamics in video generation.

In light of the above success of exploiting LLM to direct video generation in 2D space, we view LLM as a director in 3D, which differs from previous methods not only in terms of technical route but also in benefits: providing free interaction with individual concepts and flexible viewpoint control.

## 2.2 Compositional 3D Generation

Generating 3D assets from textual prompt has garnered significant attention owing to its promising applications in various fields such as AR [41], VR [42], and autonomous driving [43]. However, due to the lack of large-scale 3D data, it is challenging to apply 2D generative models to 3D directly. Therefore, building upon Dream Fields [44], DreamFusion introduced the Score Distillation Sampling (SDS) [28], a technique enhancing 3D generation by distilling 2D diffusion priors from pre-trained text-to-image generative models. Motivated by the success of DreamFusion [28], dedicated efforts have been made to improve SDS [45–47]. Though achieving remarkable results, these methods struggle to generate scenes with multiple distinct elements. To mitigate this issue, several techniques was proposed to guide 3D generation with additional conditions like layout priors, which we refer to as compositional 3D generation [48–50]. However, these works still focus on static compositional 3D generation and lack visual dynamic modeling.

Recently, two concurrent works Comp4D [51] and TC4D [52] also achieved compositional 4D generation (*i.e.*, dynamic 3D generation). However, they only considered composition between objects, and the trajectory of these methods is either formulated by kinematics-based equations or pre-defined by users. Differently, we explore 3D-aware video generation with integrated 3D scenes and compose various concepts with priors from both LLM and 2D diffusion models.

# 3 Preliminaries

## 3.1 3D Gaussian Splatting

3D Gaussian Splatting (3DGS) [25] has been attracting a lot of interest for novel view synthesis, due to its photorealistic visual quality and real-time rendering. 3DGS utilizes a set of anisotropic ellipsoids (*i.e.*, 3D Gaussians) to encode 3D properties, in which each Gaussian is parameterized by position $\mu \in \mathbb{R}^3$, covariance $\mathbf{\Sigma} \in \mathbb{R}^{3\times3}$ (obtained from scale $\mathbf{s} \in \mathbb{R}^3$ and rotation $\mathbf{r} \in \mathbb{R}^3$), opacity $\alpha \in \mathbb{R}$, and color $\mathbf{c} \in \mathbb{R}^3$.

To render a novel view, 3DGS adopts a tile-based rasterization, where 3D Gaussians are projected onto the image plane as 2D Gaussians. The final color $\mathbf{c}(\mathbf{p})$ of pixel $\mathbf{p}$ is denoted as:

$$\mathbf{c}(\mathbf{p}) = \sum \hat{\mathbf{c}}\hat{\sigma} \prod (1 - \hat{\sigma}), \tag{1}$$

where $\hat{\mathbf{c}}$ and $\hat{\sigma}$ represent the individual color and opacity values of a series of 2D Gaussians contributing to this pixel. 3DGS are then optimized using L1 loss and SSIM [53] loss in a per-view optimization manner. Thanks to the nature of modeling 3D scenes explicitly, optimized 3D Gaussians can be easily controlled and edited.

## 3.2 Score Distillation Sampling

Different from text-to-image generation which benefits from a large number of text-image pairs available, text-to-3D generation suffers from a severe lack of data. To mitigate this issue, Score Distillation Sampling (SDS) [28] was proposed to distill generative priors from pretrained diffusion-based text-to-image models $\phi$. Specifically, for a 3D representation parameterized by $\theta$, SDS is served as a way to measure the similarity between the rendered images $x = g(\theta)$ and the given textual prompts $y$, where $g$ represents the rendering operation. As a result, the gradients used to update $\theta$ are computed as follows:

$$\nabla_\theta \mathcal{L}_{SDS}(\phi, x = g(\theta)) = \mathbb{E}_{t,\epsilon}[w(t)(\hat{\epsilon}_\phi(x_t; y, t) - \epsilon)], \tag{2}$$

where $t$ is the noise level, $\epsilon$ is the ground-truth noise, $w(t)$ is a weighting function, $\hat{\epsilon}_\phi$ is the estimated noise given noised images $x_t$ with text embeddings $y$. Please refer to DreamFusion [28] for details.

# 4 Method

**Overview.** To achieve text-guided compositional 3D-aware video generation (C3V), we regard LLM as director and 3D as structural representation. To this end, our method consists of three stages. To begin with, we utilize LLMs to decompose the input textual prompts into three sub-prompts, each

Figure 1: Illustration of our method. It consists of three stages: 1) The input textual prompt is decomposed into individual concepts by the LLM. Then we generate each concept in the form of 3D with the corresponding pre-trained expert model (*left* & Sec. 4.1). 2) We leverage knowledge in multi-modal LLM to estimate the 2D trajectory of objects step-by-step (*middle* & Sec. 4.2). 3) After lifting the estimated 2D trajectory into 3D as initialization, we refine the scales, locations, and rotations of objects within the 3D scene using 2D diffusion priors (*right* & Sec. 4.3).

of which provides a description for generating a corresponding concept (*i.e.*, scene, object, motion, etc.) respectively (Sec. 4.1). Subsequently, we leverage multi-modal LLM to obtain coarse-grained scales and trajectories for each animatable object (Sec. 4.2). Finally, we employ 2D diffusion priors to refine the objects' location, scale, and rotation for a fine-grained composition (Sec. 4.3).

## 4.1 Task Decomposition with LLM

**Task Instructions.** Given a textual prompt, we invoke LLM (*e.g.*, GPT-4V [32]) to decompose it into several sub-prompts. Each sub-prompt describes an individual concept such as the scene, object, and motion. Specifically, for an input prompt $y$, we query LLM with the instruction like: "*Please decompose this prompt into several sub-prompts, each describing the scene, objects in the scene, and the objects' motion.*", from which we obtain the corresponding sub-prompts.

**3D Representation.** After obtaining the sub-prompt for each concept, we aim to generate its corresponding 3D representations using the pre-trained expert models. In this work, we build structural representation on 3DGS [25], which is an explicit form and therefore flexible enough to compose or animate. Concerning concepts like motion, our framework can generalize to arbitrary animatable 3D Gaussian-based objects. For simplicity, we take human motion as an instantiation because it is general for various scenarios. In order to obtain diverse human motions, we take a retrieval-augmented approach [54] to acquire motion in the form of SMPL-X parameters [55] from large motion libraries [56] according to the motion-related sub-prompt.

**Instantiation.** To illustrate the scheme formally, consider the following example. We have sub-prompts $y_1, y_2$ and $y_3$ that describe scene, object, and motion respectively. Additionally, we have corresponding pre-trained text-guided expert models $\phi_1, \phi_2$, and $\phi_3$ that are selected by the LLM. The concept generation can be formulated as follows:

$$G_1 = \phi_1(y_1), \ G_2 = \phi_2(y_2, M), \ M = \phi_3(y_3), \tag{3}$$

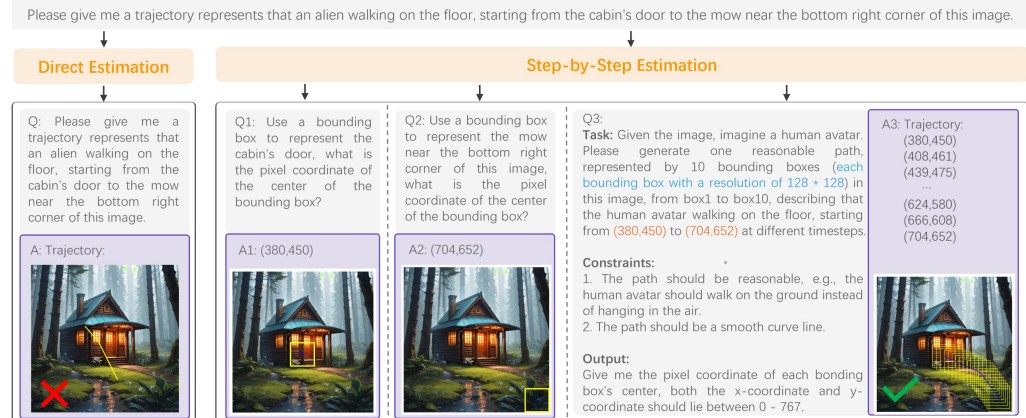

Figure 2: Illustration of coarse-grained trajectory generation with LLM. Instead of querying multi-modal LLM to estimate dynamic trajectory directly, we generate trajectory in a step-by-step manner: estimating the locations of starting and ending points first, then reasoning the path between them.

where $G_1$ and $G_2$ represent the 3D Gaussians, and $M$ means the motions used to drive $G_2$. In the following sections, we will provide details on the composition of the generated concepts.

## 4.2 Coarse-grained Trajectory Generation with LLM

Given the generated concepts, we aim to compose them into a dynamic 3D representation to render videos that align with the input textual prompt. Achieving this requires scales and trajectories of the objects to indicate their sizes and locations within the scene. To this end, we propose to leverage knowledge encoded in multi-modal LLMs (*i.e.*, GPT-4V [32]) to provide priors.

For the scale of the object, we find that directly querying GPT-4V with the input prompt and rendered scene image can yield a reasonable estimation of its resolution ($H_{2D}$ and $W_{2D}$). However, this is not the case for trajectory estimation. As demonstrated in Fig. 2, directly querying GPT-4V for trajectory will lead to a result that deviates conspicuously from common sense. Based on this observation, we conclude two issues: 1) it is too difficult for GPT-4V to generate the trajectory within a single query, as it lacks priors on visual dynamics; 2) since GPT-4V is trained to generate text, it has limitations on imagining visual content.

To mitigate this, we introduce two simple yet effective techniques. 1) Although GPT-4V lacks visual knowledge of the object, we can alleviate this by representing the object as a bounding box with the estimated resolution. 2) We follow a step-by-step reasoning philosophy [27] and propose to let GPT-4V estimate the locations of starting and ending points first, then reason the path between them.

Overall, we can formulate the above process as follows:

$$\{L_p^i\}_{i=1}^N = \Phi(y_p, I, S, L_s, L_e),$$
$$S = \Phi(y, I), L_s = \Phi(y_s, I), L_e = \Phi(y_e, I), \quad (4)$$

where $\Phi$ represents the multi-modal LLM (*i.e.*, GPT-4V), $I$ denotes the rendered scene image, $S$ represents the estimated scale of the object given textual prompt $y$ and $I$, $L_s$ and $L_e$ represent the locations of starting and ending points respectively, $\{L_p^i\}_{i=1}^N$ represent $N$ locations indicating the path between $L_s$ and $L_e$. Notably, all locations (*i.e.*, $L_s$, $L_e$, $\{L_p^i\}_{i=1}^N$ ) are represented by 2D pixel coordinates on $I$.

## 4.3 Fine-grained Composition with 2D Diffusion Priors

**Lift Trajectory from 2D to 3D.** In Sec. 4.2, we obtain the 2D pixel coordinates $L_p^i = (p_x^i, p_y^i)$ of the estimated trajectory. However, 2D trajectory is not enough for composition in 3D space. Therefore, we convert it into corresponding 3D world coordinate $L_{3D}^i = (x^i, y^i, z^i)$. Specifically, we first predict the depth map $D$ of the rendered scene image with a monocular depth estimator [57].

Then, we use the depth value of the center point of the lower boundary of the bounding box as the trajectory's depth. As a result, we can transform 2D trajectory into 3D:

$$(x^i, y^i, z^i, 1)^T = R^{-1}K^{-1}[(p_x^i + \frac{H_{2D}}{2}, p_y^i, 1)^T \cdot D(p_x^i + \frac{H_{2D}}{2}, p_y^i)] - (\frac{H_{3D}}{2}, 0, 0, 0)^T, \quad (5)$$

where $R$ and $K$ represent camera extrinsic and intrinsic respectively, $H_{2D}$ and $W_{2D}$ represent the resolution of the 2D bounding box. $H_{3D}$ represent the actual height of the 3D bounding box of this object within the scene.

**Composition Refinement with 2D Diffusion Priors.** With the lifted 3D trajectory, we then integrate the object into the scene. However, the trajectory estimated by LLM is still rough and may not obey natural image distribution. To address this, we propose to further refine the object's scale, location, and rotation by distilling generative priors from pre-trained image diffusion models [7] into 3D space. Specifically, we treat the parameters for these attributes as optimizable variables and use SDS (Eq. 2) to improve the fidelity of rendered images. As a result, scale refinement can be formulated as follows:

$$\nabla_{\hat{S}} \mathcal{L}_{SDS}^{Scale} = \mathbb{E}_{t,\epsilon}[w(t)(\hat{\epsilon}_\phi(x_t(L_{3D}^1, (S + \sigma(\hat{S}) \cdot \tau_s - \frac{\tau_s}{2}) \cdot G_2); y, t) - \epsilon)], \quad (6)$$

where $\hat{S}$ represents the optimizable variable, $S$ represents the scale estimated by GPT-4(V), $\sigma$ means the Sigmoid function, $\tau_s$ is a threshold, $G_2$ represents the 3D gaussians of the object, and $x_t$ is the noised 2D image given $L_{3D}^i$ and scaled $G_2$.

After obtaining a more precise scale, we then refine the locations of the estimated 3D trajectory similarly, where the location refinement is denoted as:

$$\nabla_{\hat{L}^i} \mathcal{L}_{SDS}^{Location} = \mathbb{E}_{t,\epsilon}[w(t)(\hat{\epsilon}_\phi(x_t(L_{3D}^i + \sigma(\hat{L}^i) \cdot \tau_L - \frac{\tau_L}{2}, (S + \sigma(\hat{S}) \cdot \tau_s - \frac{\tau_s}{2}) \cdot G_2); y, t) - \epsilon)], \quad (7)$$

where $\hat{L}^i$ represents the optimizable variable, $\tau_L$ is a threshold.

For the rotation of the object at different timesteps, we can directly compute the corresponding rotation matrix, based on the assumption that the object at the current time step should face the location of the object at the next time step. As a result, the rotation matrix $\hat{R}^i$ at time step $i$ can be computed using the following equation:

$$\hat{R}^i = \begin{bmatrix} tx^2 + c & txy - zs & txz + ys \\ txy + zs & ty^2 + c & tyz - xs \\ txz - ys & tyz + xs & tz^2 + c \end{bmatrix}, \quad (8)$$
$$t = 1 - c, c = \cos(\theta), s = \sin(\theta), \mathbf{u} = (x, y, z)^T$$

where $\theta$ and $\mathbf{u}$ represent the rotation angle and axis obtained through the cross product of $(L_{3D}^{i+1} + \sigma(L^{\hat{i}+1}) \cdot \tau_L - L_{3D}^i - \sigma(\hat{L}^i) \cdot \tau_L)$ and $(0, 0, 1)^T$.

**Inference.** After obtaining individual concepts in the form of 3D and the optimized parameters that indicate how to compose various concepts, we can render the 3D representation into 2D video with flexible camera control in real time [25].

## 5 Experiments

In this section, we instantiate C3V with three concepts: scene, humanoid object, and human motion, to generate 3D-aware video from text. We compare our proposed method with state-of-the-art text-to-4D models (4D-FY [58]), compositional 4D generation models (Comp4D [51]) and text-to-video models (VideoCrafter2 [59]). Videos are available on our anonymous project page.

**Implementation Details.** We use LucidDreamer [60], HumanGaussian [61] and Motion-X [56] to generate 3D scenes, humanoid objects and motions respectively. To realize SDS, we utilize Stable Diffusion [7] as the image diffusion model. All the videos of our proposed method are rendered at a resolution of $512 \times 512$ in real time. Please refer to the appendix for more details.

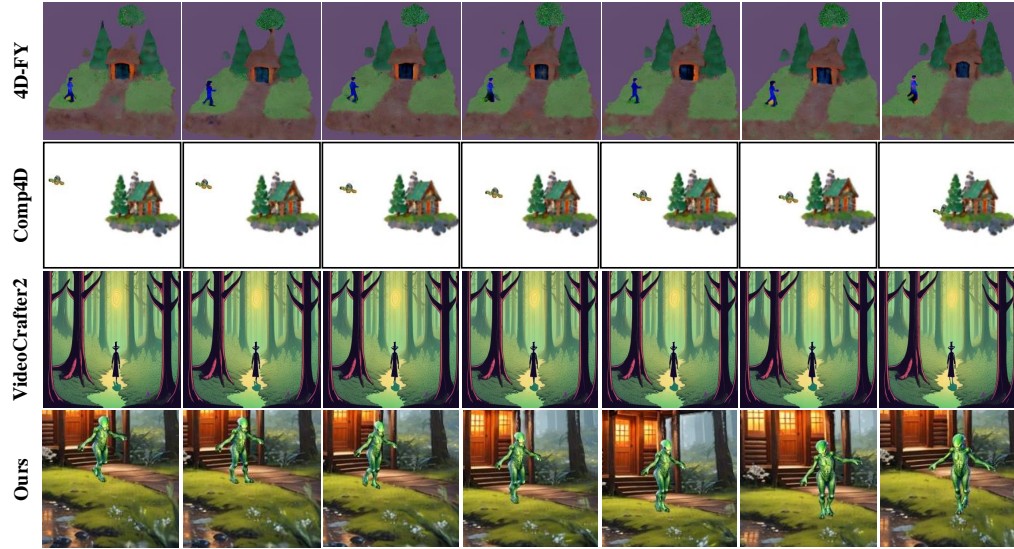

(a) Text prompt: "*In a Magician's magical cabin alone in a serene forest, an alien walking on the floor, starting from the cabin's door to the mow near the bottom right corner of this image*".

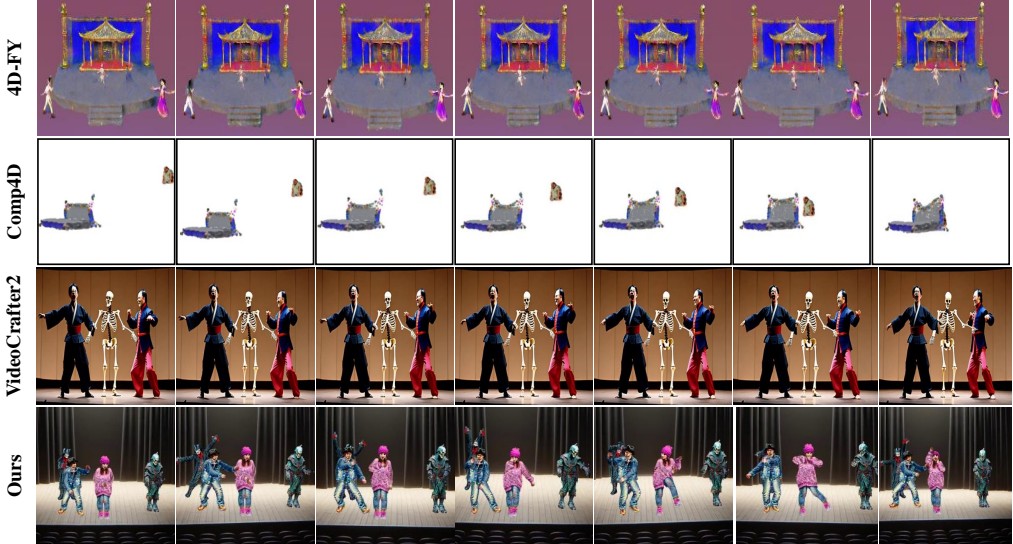

(b) Text prompt: "*Four characters stood on the stage. In front of the stage, a man and a woman are performing Kung Fu and dancing respectively. On the right side of the stage, a skeleton man is dancing, and behind them, a clown is performing*".

Figure 3: Qualitative comparisons with baselines. When prompting complex queries, the baseline methods fail to follow the queries in terms of the number of objects and the corresponding motion. In contrast, our method excels in yielding both diverse motion and high visual quality.

**Metrics.** Following Comp4D [51], we choose Q-Align [62] as the referee to measure the quality and aesthetics of the video. The Q-Align score is a number ranging from 1 (worst) to 5 (best) where a higher score indicates a better performance. We also report the CLIP score [63] to measure the alignment between the generated videos and the input texts.

## 5.1 Comparison with Competitors

In Fig. 3, we conduct a comparative analysis of our method against 4D-FY [58], Comp4D [51], and VideoCrafter2 [59] with the same textual prompt. It can be observed that all three baselines fail to provide diverse motion from the textual prompt, while our method excels in yielding large motion and high visual quality. For example, our scheme successfully obeys the complex query in terms of

Table 1: Quantative comparisons with competitors. Our method consistently outperforms all baseline methods in terms of both the video quality and the alignment with textual prompts.

| Metric | 4D-FY [58] | Comp4D [51] | VideoCrafter2 [59] | Ours |
|---|---|---|---|---|
| QAlign-img-quality ↑ [62] | 1.681 | 1.687 | 3.839 | **4.030** |
| QAlign-img-aesthetic↑ [62] | 1.475 | 1.258 | 3.199 | **3.471** |
| QAlign-vid-quality↑ [62] | 2.154 | 2.142 | 3.868 | **4.112** |
| QAlign-vid-aesthetic↑ [62] | 1.580 | 1.425 | 3.159 | **3.723** |
| CLIP Score↑ [63] | 30.47 | 27.50 | 35.20 | **38.36** |

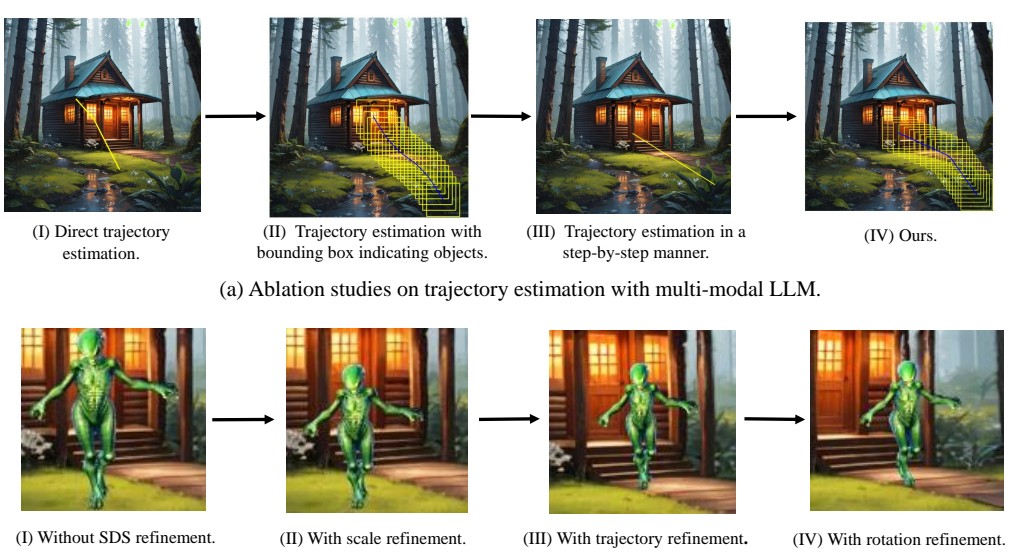

(I) Direct trajectory estimation. (II) Trajectory estimation with bounding box indicating objects. (III) Trajectory estimation in a step-by-step manner. (IV) Ours.

(a) Ablation studies on trajectory estimation with multi-modal LLM.

(I) Without SDS refinement. (II) With scale refinement. (III) With trajectory refinement. (IV) With rotation refinement.

(b) Ablation studies on composition with 2D diffusion models.

Figure 4: Ablation studies on framework design. Each ablation is prompted with the same text.

the number of objects and the corresponding motion. In addition, since 4D-FY and Comp4D focus on object-centric generation, they fail to generate videos with natural backgrounds. In Tab. 3, we perform quantitative comparisons by utilizing Q-Align Score [62] and CLIP Score [63] to assess the quality of generated videos. Our method consistently outperforms the baseline models in terms of both the video quality and the alignment with textual prompts. More results are available in the appendix.

## 5.2 Ablation Studies

**Ablations on Trajectory Estimation with Multi-modal LLM.** As shown in Fig. 4(a)(I), a direct prompt of GPT-4V will lead to obvious unsatisfactory trajectory estimation. When only depending on bounding boxes to indicate the location of objects within the scene (Fig. 4(a)(II)), though a roughly better trajectory can be achieved, it still leads to unreasonable results, such as several floating bounding boxes. Similarly, using only the step-by-step estimation strategy described in Sec. 4.2 typically results in a trajectory that is merely a simple straight line connecting the starting and ending points (Fig. 4(a)(III)). With both of the two techniques, we can achieve the best performance, with a more reasonable and smooth trajectory (Fig. 4(a)(IV)).

**Ablations on Composition with 2D Diffusion Models.** To figure out whether it is necessary to conduct fine-grained composition with 2D generative priors, we gradually refine the scales, locations, and rotations with SDS and visualize the results in Fig. 4(b). All results are generated with the same textual prompt: "*An alien walking on the floor in front of the cabin's door.*". It shows that when we optimize the attributes with SDS, we can obtain consistently improved performance with a reasonable scale (Fig. 4(b)(II), accurate locations that are aligned with the input prompt (Fig. 4(b)(III), and orientation that accords with common sense (Fig. 4(b)(IV)).

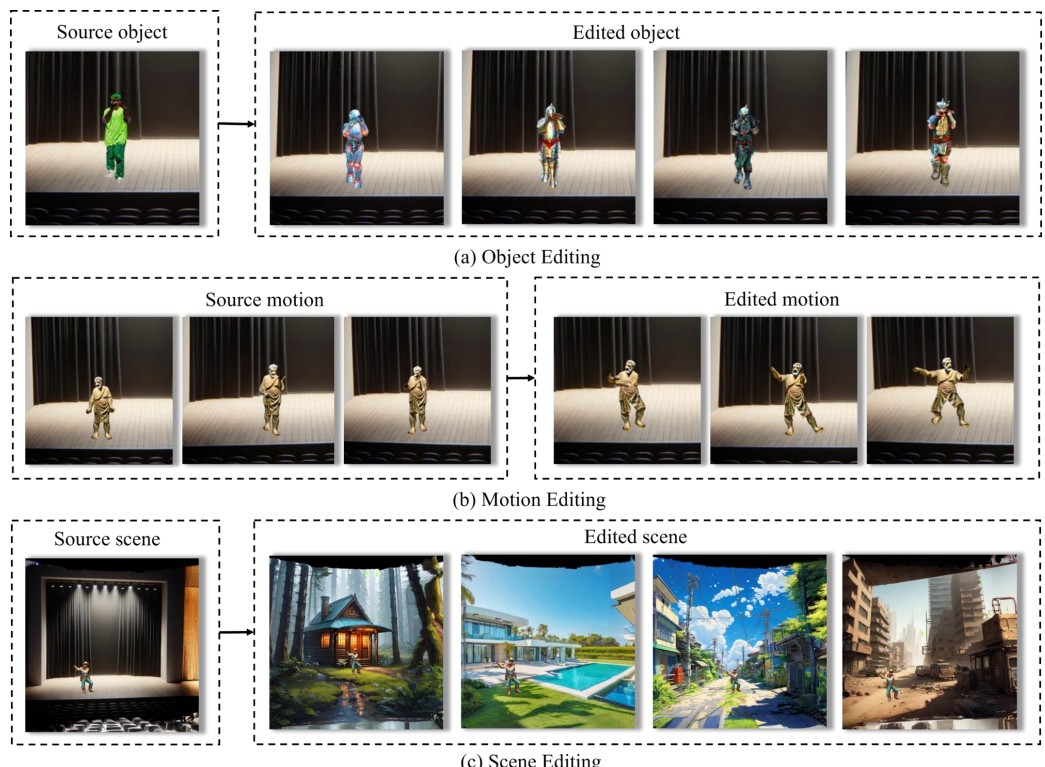

(a) Object Editing

(b) Motion Editing

(c) Scene Editing

Figure 5: Our method offers flexible control of individual concepts. We demonstrate this by editing different concepts: the appearance and motion of the actors, and the scenes.

Table 2: Quantative comparisons of ablation studies on trajectory estimation with multi-modal LLM.

| Methods | Direct Estimation. | Estimation using bounding box. | Step-by-step estimation. | Ours |
|---|---|---|---|---|
| QAlign-img-quality ↑ [62] | 2.056 | 2.894 | 3.752 | **4.030** |
| QAlign-img-aesthetic ↑ [62] | 1.568 | 2.156 | 3.047 | **3.471** |
| QAlign-vid-quality ↑ [62] | 2.178 | 3.043 | 3.904 | **4.112** |
| QAlign-vid-aesthetic ↑ [62] | 1.680 | 2.346 | 3.342 | **3.723** |
| CLIP Score ↑ [63] | 25.68 | 29.84 | 36.73 | **38.36** |

## 5.3 Applications on Controllable Generation

Due to our underlying 3D structural representation, our scheme has the natural merits of editing individual concepts. We illustrate this character in Fig. 5 by editing three different concepts: the appearance and motion of the actors, and the scenes. For the appearance and motion of the actor, we can seamlessly replace them in a zero-shot manner according to the textual prompt (Fig.5(a)(b)), while this is still challenging for implicit models [64, 65]. For scene editing, to ensure a smooth composition of objects within the target scene, we re-estimate the trajectory of the objects given the target scene. Kindly refer to appendix for more results.

## 6 Conclusion

In this paper, we present a novel paradigm for 3D-aware video generation by conceptualizing videos as compositions of independent concepts represented in 3D space. To this end, we leverage LLM as director to decompose the input textual prompts into individual concepts and then invoke pre-trained expert models to generate them separately. To compose various concepts, we first prompt multi-modal LLM in a step-by-step manner to provide coarse guidance on the scale and trajectory of objects,

Table 3: Quantative comparisons of ablation studies on composition with 2D diffusion models.

| Methods | Without SDS. | With scale refinement. | With trajectory refinement. | Ours |
|---|---|---|---|---|
| QAlign-img-quality ↑ [62] | 3.045 | 3.674 | 3.826 | **4.030** |
| QAlign-img-aesthetic ↑ [62] | 2.752 | 3.046 | 3.341 | **3.471** |
| QAlign-vid-quality ↑ [62] | 3.129 | 3.794 | 3.983 | **4.112** |
| QAlign-vid-aesthetic ↑ [62] | 2.704 | 3.468 | 3.603 | **3.723** |
| CLIP Score ↑ [63] | 31.35 | 35.27 | 37.04 | **38.36** |

then refine the composition with 2D generative priors. We verify our scheme in different scenarios, demonstrating its superiority over the baseline methods.

**Limitations and Future Works.**  Although we demonstrate promising results in 3D-aware video generation, there still are limitations to be improved in the future. First, our framework is instantiated with limited concepts in this work, *i.e.*, scene, humanoid object, and human motion. It is exciting to generalize the framework to more concepts like animals, vehicles, etc. Second, the composition between concepts is conducted with priors from LLM and 2D diffusion priors in our method. However, it is still interesting to introduce physically grounded dynamics into 3D representation [66]. Third, though our method is naturally suitable for maintaining the consistency of actors across different scenes, it still needs further exploration on long video generation with multiple scenes, *e.g.*, a full-length film.

**Ethics Statement.**  C3V is exclusively a research initiative with no current plans for product integration or public access. We are committed to adhering to Microsoft AI principles during the ongoing development of our models. The model is trained on AI-generated content, which has been thoroughly reviewed to ensure that they do not include personally identifiable information or offensive content. Nonetheless, as these generated data are sourced from the Internet, there may still be inherent biases. To address this, we have implemented a rigorous filtering process on the data to minimize the potential for the model to generate inappropriate content.

**Acknowledgement.**  This work was supported in part by NSFC under Grant 62371434, 62021001.

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

# A   Implementation Details

## A.1   Experimental Settings

During the process of multi-modal LLMs-based trajectory estimation, we use 20 locations by default to indicate the trajectory between the starting point and the ending point, (*i.e.*, $N = 20$ in Eq. 4), where the path between adjacent locations is assumed as a straight line. For scale refinement (Eq. 6), $\tau_s$ is set to 0.1. For location refinement (Eq. 7), we apply it to refine all the twenty locations. The training iterations for each location is 1000 and $\tau_L$ is set to 0.1. All experiments are conducted using a single NVIDIA A100 GPU.

## A.2   Pre-trained Expert Models

**LucidDreamer [60].**   As a powerful 3D scene generation method, LucidDreamer adopts an iterative view generation strategy, where a series of views are dreamed and aligned via depth-warping based inpainting networks. After obtaining these multiview-consistent images, 3D gaussians are optimized to construct a high-quality 3D scene, by means of typical training pipeline of 3DGS.

**HumanGaussian [61].**   We choose HumanGaussian as the method for human generation due to its capability of producing drivable avatars on the basis of 3DGS. Specifically, HumanGaussian starts with SMPL-X prior to densely sample Gaussians on the human mesh surface as initial center positions, followed by a texture-structure joint model and an annealed negative prompt guidance strategy to obtain high-fidelity outputs.

**Motion-X [56].**   Motion-X is a large-scale 3D expressive whole-body human motion dataset which comprises 15.6M precise 3D whole-body pose annotations (*i.e.*, SMPL-X) covering 81.1K motion sequences from massive scenes with sequence-level semantic labels. By calculating the similarity of text embeddings between the motion-related sub-prompt and sequence labels, we find the most matching sequence and acquire motion data in the form of SMPL-X parameters [55].

## A.3   Metrics

Given the lack of ground truth videos for specific text queries, we utilize pre-trained quality-assessment models to evaluate the generated videos and their individual frames. In line with Comp4D [51], we employ Q-Align [62] as the benchmarking tool to assess the quality and aesthetics of the videos. The Q-Align rating, which ranges from 1 (worst) to 5 (best), is considered state-of-the-art, closely aligning with human judgments across established quality assessment benchmarks. Additionally, we include the CLIP score [63] to measure the alignment between the generated videos and the input texts, where higher scores signify better alignment.

# B   Results of Different Stages

As shown in Fig. 6, Fig. 7 and Fig. 8, we provide a detailed visualization of results obtained during different stages, including results of LLM-based task decomposition, results of coarse-grained trajectory generation with GPT-4V, and results of the final rendered videos.

# C   More Results of Controllable Generation

In this section, we present additional results on controllable generation. As demonstrated in Fig. 9, Fig. 10 and Fig. 11, we can achieve fine-grained control of the target without affecting other concepts in the 3D space.

**Prompt:** In a Magician's magical cabin alone in a serene forest, an alien walking on the floor, starting from the cabin's door to the mow near the bottom right corner of this image.

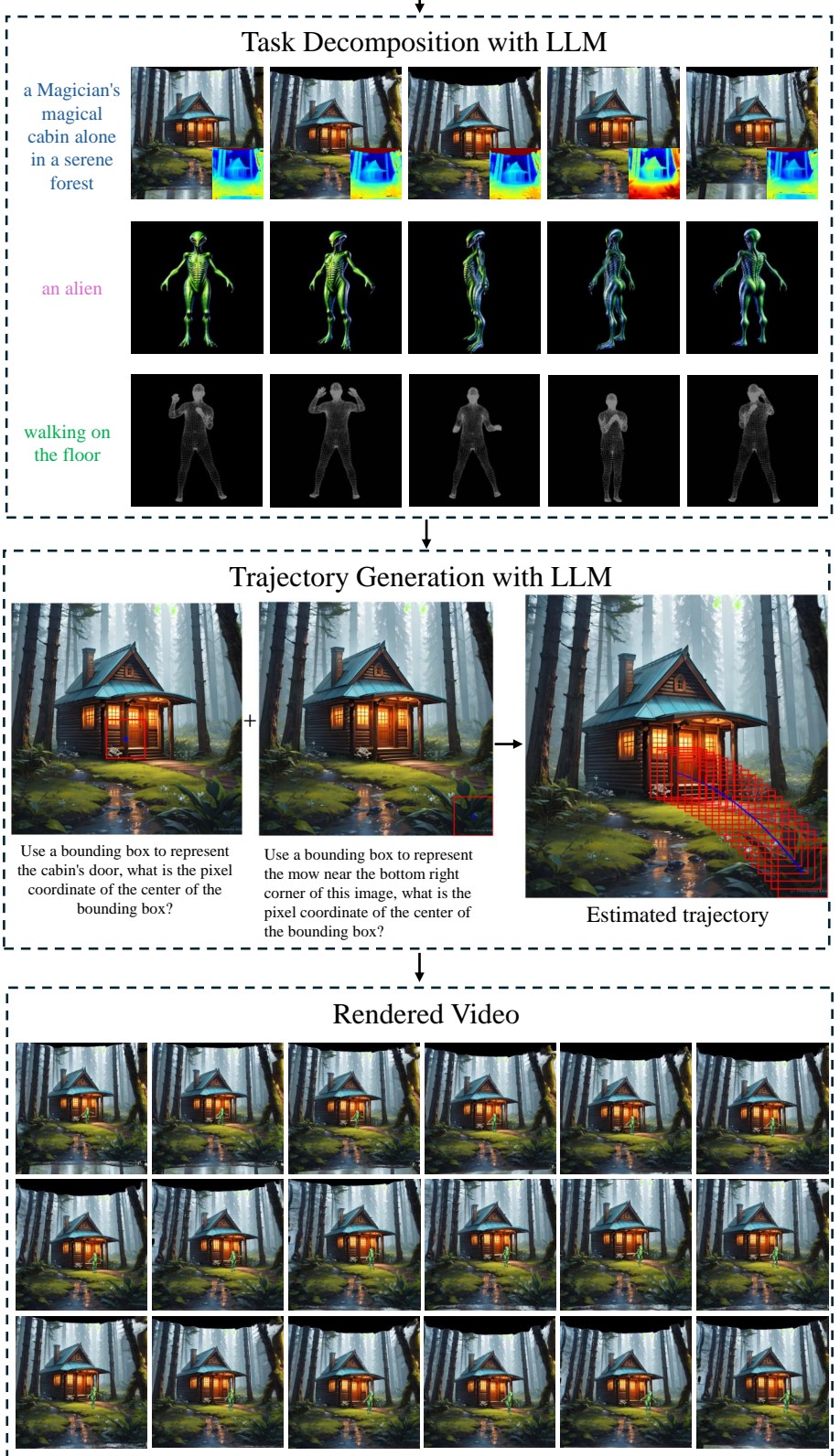

Figure 6: Results of different stages given the textual prompt: "*In a Magician's magical cabin alone in a serene forest, an alien walking on the floor, starting from the cabin's door to the mow near the bottom right corner of this image.*".

**Prompt:** Inside a cozy livingroom in Christmas, a astrologer performing ballet on the floor, starting from wooden floor behind the red armchair near the bottom left of this image to the sofa in the bottom right corner of this image.

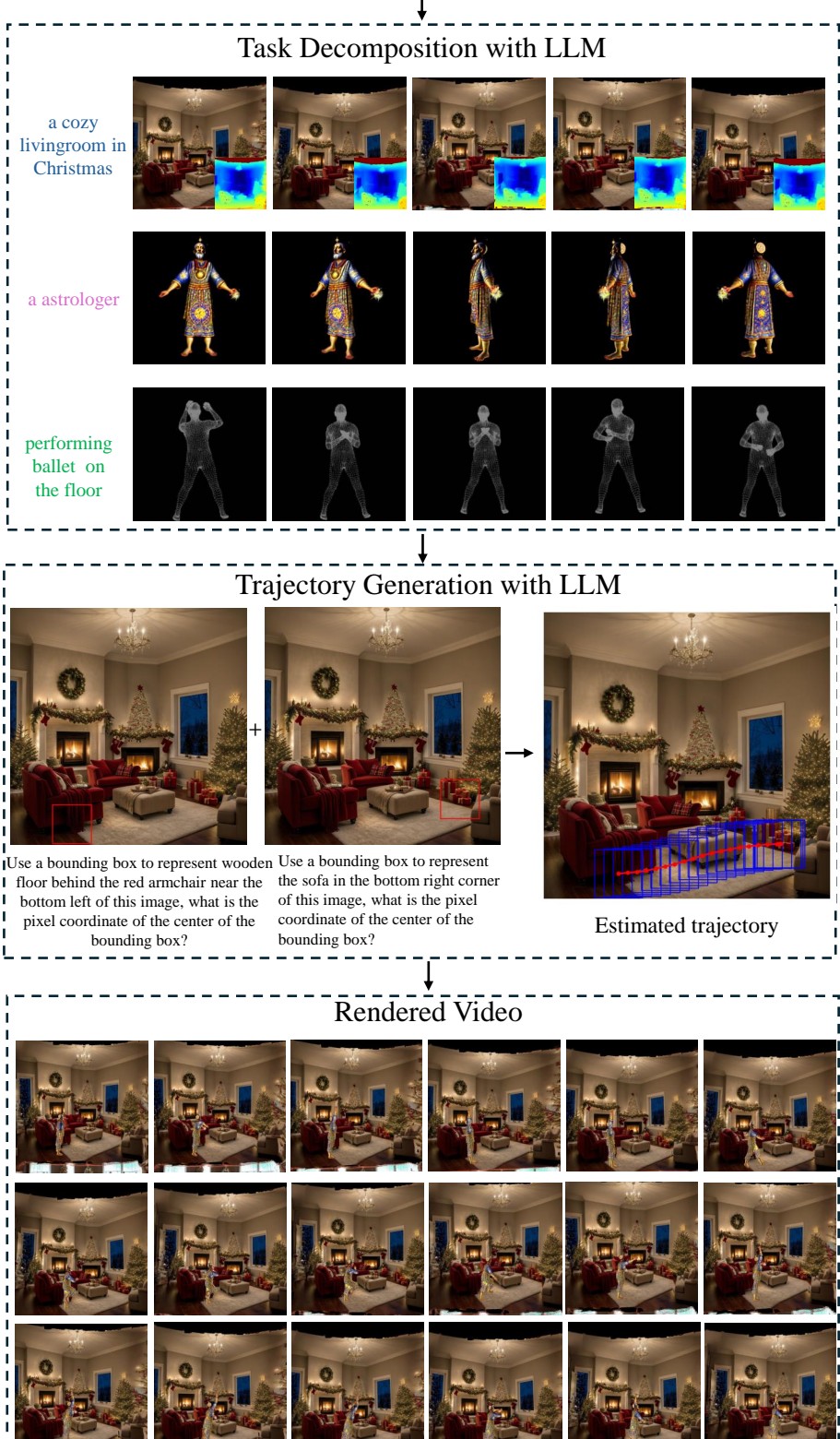

Figure 7: Results of different stages given the textual prompt: "*Inside a cozy livingroom in Christmas, a astrologer performing ballet on the floor, starting from wooden floor behind the red armchair near the bottom left of this image to the sofa in the bottom right corner of this image.*".

**Prompt:** On a simple stage, a man with a black fedora and a denim jacket and a woman wearing ski clothes are performing Kungfu and dancing respectively, on the left side and right side of this stage.

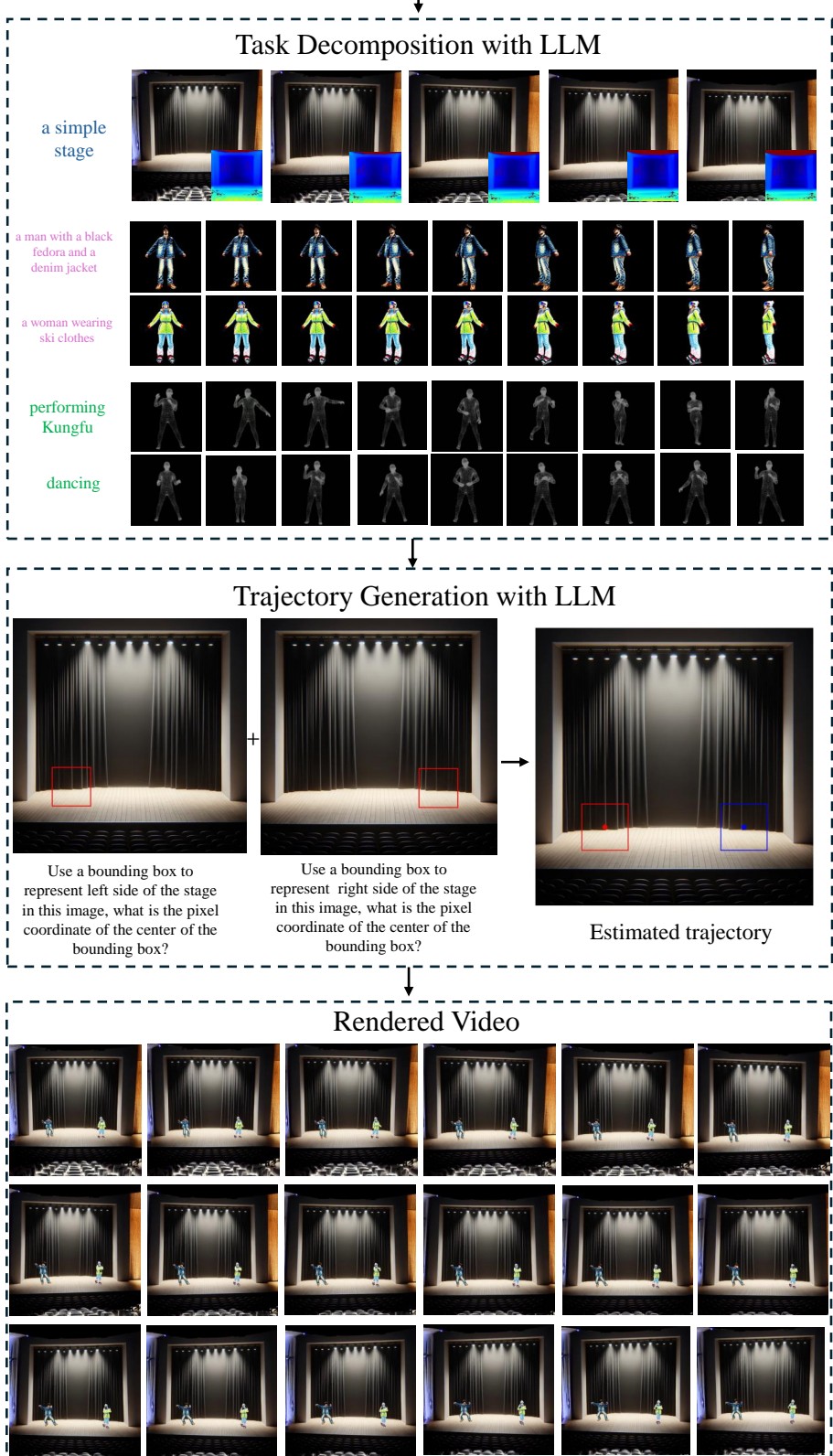

Figure 8: Results of different stages given the textual prompt: "*On a simple stage, a man with a black fedora and a denim jacket and a woman wearing ski clothes are performing Kungfu and dancing respectively, on the left side and right side of this stage.*".

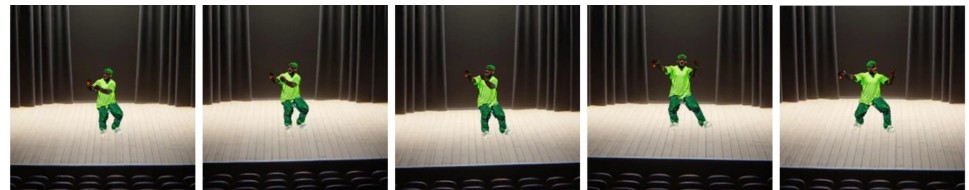

**(a) Prompt: A black man wearing a green t-shirt playing Kungfu on the stage.**

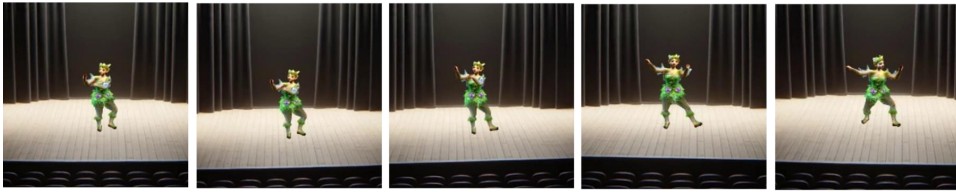

**(b) Prompt: Turn the character into a fairy.**

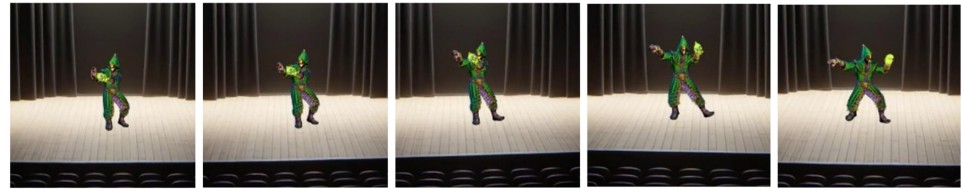

**(c) Prompt: Turn the character into a warlock.**

Figure 9: Results of actor's appearance editing.

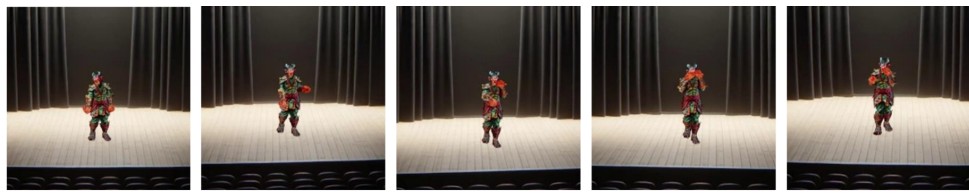

**(a) Prompt: A demon slayer performing a han and tang dance on the stage.**

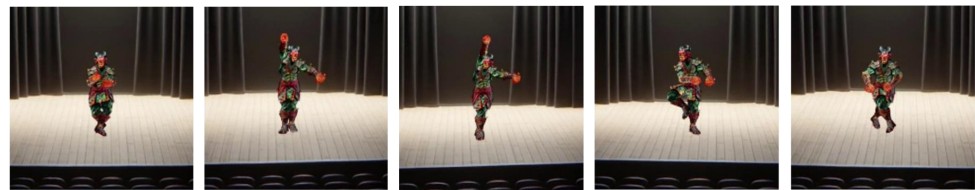

**(b) Prompt: Turn the motion of the character into a ballet.**

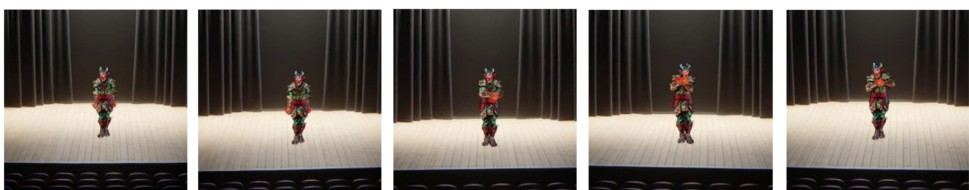

**(c) Prompt: Turn the motion of the character into a kin kong exotic dance.**

Figure 10: Results of actor's motion editing

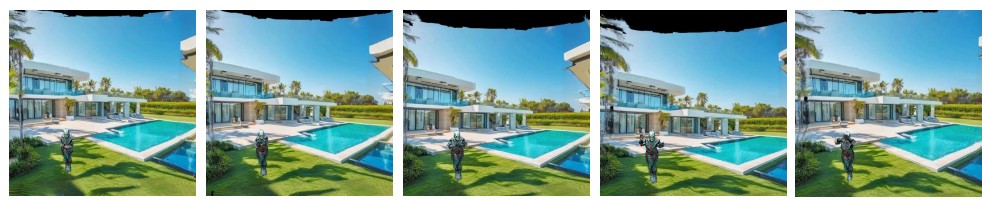

**(a) Prompt: A revenant dancing Daily Lovedive in a ultra-modern mega villa by the sea with swimming pool.**

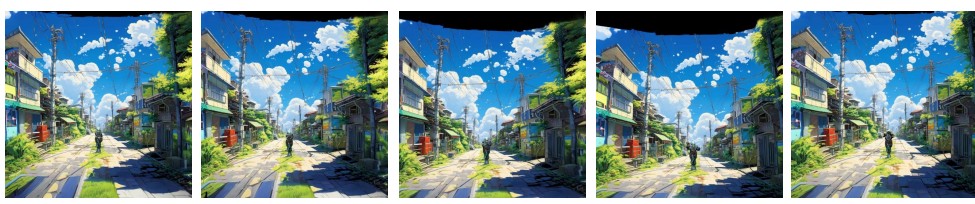

**(b) Prompt: Turn the scene into a long anime-style road with anime-blocks and little anime-grass.**

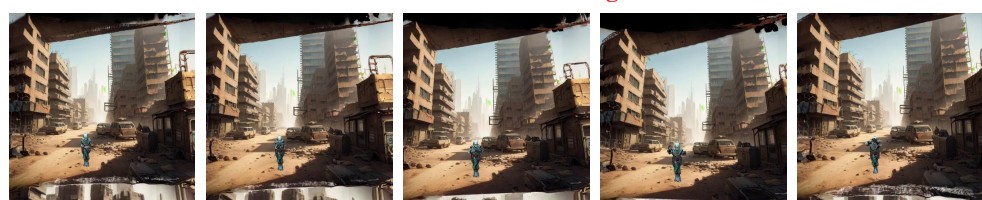

**(c) Prompt: Turn the character into a postapocalyptic city in desert.**

Figure 11: Results of scene editing.

