# OpenReview forum: "Compositional 3D-aware Video Generation with LLM Director"
_NeurIPS.cc/2024/Conference — NeurIPS 2024 poster_

### Official Review · Reviewer_BLtC · 2024-07-12

**Soundness:** 3
**Presentation:** 3
**Contribution:** 3
**Rating:** 5
**Confidence:** 4

**Summary:**

This paper presents an LLM-involved three-stage pipeline for text-guided compositional 3D-aware video generation. In the first stage, an LLM is employed as director to decompose input textual prompts into sub-prompts including scene, object and motion. Subsequently, it leverages multi-modal LLM to make an initial estimation about scales and trajectories for each object. Moreover, 2D diffusion priors are further leveraged to refine the 3D generation results with SDS loss. Extensive experiments demonstrate that the effectiveness of the proposed method in 3D-aware high-fidelity video generation.

**Strengths:**

1.The idea of decompose the generation task into scene, object and its motion in a more organized way with LLM is interesting.
2.The method of generating trajectories following a step-by-step manner is a reasonable solution and is validated in the experiments.
3.Using SDS loss to distill generative priors from pretrained LDM/SD models helps refine the quality of generation and is validated in the ablation study.
4.With LLM as director and 3D as structural representation, the proposed method is able to generate 3D-aware videos with diverse motion and high visual quality.

**Weaknesses:**

1.How to choose pretrained expert models? How would those models influence generation performance?
2.Since the proposed method adopts the divide-and-conquer strategy, I wonder how it perform in case of complex scene with multiple objects, especially with interaction among objects.

**Questions:**

1.Rendering 2D video from 3D representation is real-time as said in the paper, what about the time spent on 3D representation generation and trajectory generation ?
2.How is the influence of selection	 of pretrained expert models and open-source LLM/MLLM as director?

**Limitations:**

Yes, limitations have been discussed in the paper.

---

> ### Author Rebuttal · Authors · 2024-08-06
>
> Dear Reviewer BLtC, thank you for taking the time to review our work and for your positive and insightful feedback. We are pleased to hear that you found our idea interesting and that it demonstrated improved performance. We hope the following comments address your concerns.
>
> **Q: Criteria for choosing pretrained expert models.**
>
> A: In this paper, for scene and object generation, we choose models based on 3D Gaussians considering their explicit structure, which is beneficial for composition and editing. For motion generation, we use models capable of leveraging the generated motions to drive the object's 3D Gaussians.
>
> **Q: Performance with multiple objects.**
>
> A: As shown in Fig. 3(b) in the paper and the results in the attached PDF file, we can still achieve satisfactory results when generating videos with multiple objects and complex scenes. By decomposing the video generation task into several sub-tasks, we can pre-generate multiple objects' motions and then compose them into the same scene to produce a coherent video. In the future, we plan to incorporate more physics-based priors to generate more complex interactions.
>
> **Q: Time spent on 3D generation and trajectory generation.**
>
> A: Since we use pre-trained expert models to generate corresponding 3D representations, the time needed depends on the model itself. Specifically, for trajectory estimation, it takes about one minute for GPT-4V to make an inference. For scene generation using LucidDreamer, it takes about 15 minutes to generate a scene. For object generation with HumanGaussian, it takes approximately one and a half hours to generate an object. However, as noted by Reviewer 6K4s, the performance can be improved by leveraging more powerful expert models, such as [1] for scene generation and [2] for object generation, which will significantly accelerate the generation process. We will explore this in future works.
>
> [1] InstantSplat: Unbounded Sparse-view Pose-free Gaussian Splatting in 40 Seconds
>
> [2] Animatable 3D Gaussian: Fast and High-Quality Reconstruction of Multiple Human Avatars
>
> **Q: Influence on choosing different expert models.**
>
> A: According to our criteria for selecting pretrained expert models, when using models based on 3D Gaussians, no significant performance gap is observed. For trajectory estimation, since we follow a step-by-step approach, we believe that reasonable trajectories can also be generated by open-source MLLMs. We will validate this in future works.

---

> > ### Comment · Reviewer_BLtC · 2024-08-14
> > **response**
> >
> > I appreciate the efforts of authors in responding my comments. My concerns have been partially addressed, but the influence of expert models and LLMs/MLLMs are not investigated sufficiently. After reading other reviewers' comments, I also think the quality of the generation should be further improved. Hence I would change my score to 5 borderline accept.

---

> > > ### Author Response · Authors · 2024-08-14
> > > **We hope that our response addresses your misunderstanding.**
> > >
> > > Dear Reviewer BLtC,
> > >
> > > Thanks for your reply! We would like to thank you for your involvement and are happy to see that our response has partially addressed your concerns.
> > >
> > > **Q: The influence of different expert models and LLMs/MLLMs.**
> > >
> > > We aim at compositing 3D scenes, objects and motion with LLM director in this work. Therefore, **our focus is on how to guide the composition with prior knowledge from LLM and diffusion models** (by generating transformation and trajectory step-by-step and refining them with 2D diffusion priors). To achieve this goal, we instantiate our idea with the state-of-the-art expert models and LLMs. We have also deeply investigated the composition in Table 1 of the paper, as well as Table 1 and Table 2 in our response to Reviewer 7svJ, which is acknowledged by Reviewer 7svJ.
> > >
> > > We truly appreciate the suggestions on the investigation of various expert models and LLMs/MLLMs, which may further enrich our work. However, as it is not the focus in this work, we are disheartened that this point may have an impact on your evaluation.
> > >
> > > **Q: The quality of the generation.**
> > >
> > > As detailed in our responses to Reviewer 7svJ and as shown in Table 1 of the paper, we generated **400 videos** featuring diverse scenes, objects, and motions. The average scores for both CLIP Score and Q-Align Score are presented below:
> > >
> > > **Table. 1**: Quantitative Comparisons with Competitors.
> > >
> > > | Metric                    | 4D-FY | Comp4D | VideoCrafter2 | Ours  |
> > > |---------------------------|----------------------------|----------------------------|--------------------------------------------|--------|
> > > | QAlign-img-quality $\uparrow$       | 1.681                      | 1.687                      | 3.839                                      | **4.030** |
> > > | QAlign-img-aesthetic $\uparrow$     | 1.475                      | 1.258                      | 3.199                                      | **3.471** |
> > > | QAlign-vid-quality $\uparrow$       | 2.154                      | 2.142                      | 3.868                                      | **4.112** |
> > > | QAlign-vid-aesthetic $\uparrow$     | 1.580                      | 1.425                      | 3.159                                      | **3.723** |
> > > | CLIP Score $\uparrow$   | 30.47                      | 27.50                      | 35.20                                      | **38.36** |
> > >
> > > The experimental results illustrate that our proposed method represents a significant advancement over existing state-of-the-art  / concurrent approaches, achieving notable improvements in terms of quality, aesthetics, and alignment with input text prompts.
> > >
> > > If you have any further questions, please feel free to reach out.

---

### Official Review · Reviewer_7svJ · 2024-07-12

**Soundness:** 2
**Presentation:** 3
**Contribution:** 2
**Rating:** 3
**Confidence:** 5

**Summary:**

This paper proposes a method to synthesize dynamic 3D videos, with moving objects and camera. It treats the tasks compositionally, generating the (static) background and foreground figures separately. The generation process is orchestrated by an LLM, which provides prompts to separate models that specialize in different scene components (and which are pretrained); these components each use gaussian spats allowing assembly into a single dynamic 4D scene. This compositionality also enables controllable generation, where certain scene elements are replaced depending on user input. The method is demonstrated on several text prompts, and shown to out-perform three baselines.

**Strengths:**

- The idea of treating 3D/4D video generation as a compositional task is elegant, and it is sensible to leverage existing strong domain-specific models; it is also a nice idea to use an LLM here to automatically determine a suitable sequence of domain models to apply, and how to combine them.
- The proposed pipeline is novel, and fairly natural for the task. The choice of stages/components is clearly motivated.
- The method successfully generates videos from at least two text prompts, showing somewhat plausible motion. Qualitative results from two prompts show significantly better visual quality than the selected baselines (VideoCrafter, Comp4D, 4Dfy)
- Quantitative results based on CLIP score (adherence of frames to prompt) and Q-Align again exceed the baselines
- As well as text-conditioned generation, the method also supports certain other kinds of controllability. Since the scene representation is compositional, foreground humanoids can be replaced by others, motion can be modified, and the background can be replaced. This affords a degree of precise control that is missing from 'monolithic' video generation models
- Ablation experiments were conducted, removing three components, aiming to establish their importance in the overall pipeline
- The writing is generally clear (modulo a few grammar issues); the paper is well organized.

**Weaknesses:**

- Very few qualitative examples are given (and presumably these were cherry-picked rather than random). In particular, only two text-conditioned generations are shown (same in paper and supplementary). This makes it difficult for the reader to judge the visual quality of the model outputs, which is vital for such a task
- Even in the given two examples, prompt adherence is poor, with the "skeleton man" missing, incorrect positioning ("in front" of the stage vs at the front; "in" a cabin vs outside), and unrealistic lighting (no shadowing). This is problematic given that object compositionality being guided by the LLM is claimed as a key contribution of the work
- It is unclear how many videos were used for the quantitative evaluation, nor where the set of prompts was drawn from. This makes it hard to judge the significance of these results.
- The ablation experiments are on a prompt and qualitative only. This means they are statistically meaningless, and the benefits of the different components need to be demonstrated more rigorously.
- The exact prompting strategy for the LLM is unclear, in particular the initial stage of creating the sub-tasks, and the creation of the scale/trajectory estimation prompt.
- The method is limited by the choice of 'experts' that synthesize parts of the scene (currently static background from LucidDreamer, humanoids from HumanGaussian, and humanoid motion from Motion-X). While 'delegating' generation subtasks is a neat idea, it seems that significant engineering work is required to incorporate each 'expert', and there is not a clear path to adding e.g. other dynamic object types such as quadruped animals.

**Questions:**

- How many prompts were in the evaluation set? How were these selected?
- What are the quantitative results from the ablation study?

**Limitations:**

There is adequate discussion of limitations. There is an exceedingly brief discussion of broader impacts, borderline adequate for this task.

---

> ### Author Rebuttal · Authors · 2024-08-06
>
> Dear Reviewer 7svJ, thank you for your thoughtful feedback and for looking into every detail of our work. We are pleased that you found our idea elegant, novel, clearly motivated, and well-organized. We hope the following comments address your concerns.
>
> **Q: More qualitative examples should be provided.**
>
> A: Thanks for pointing this out. In fact, the two examples in this paper were randomly selected rather than cherry-picked. In the attached PDF file, we provide additional five examples with different 3D viewpoints and corresponding depth maps, demonstrating similar performance. We will release more results in the future version of our paper.
>
> **Q: Prompt adherence is poor.**
>
> A: Compared to competitors like 4D-fy, Comp4D, and VideoCrafter2, our method significantly outperforms them in generating results that more accurately adhere to the text prompts. However, despite this improvement, some inconsistencies in detail are currently unavoidable due to the limitations of the expert models adopted. For example, the generated "skeleton man" from HumanGaussian is indeed the actor on the right side of the stage (Fig. 3(b) in the paper), and it will be difficult for the LLM to estimate a trajectory "inside the cabin" when the 2D image it relies on represents scenes "outside the cabin." Additionally, this paper does not consider factors such as illumination, as this is the first work towards compositional 3D-aware video generation, and we only consider several basic properties. As mentioned by Reviewer 6K4s, our pipeline can be easily improved by incorporating better modules for each component, and we plan to address this in future works.
>
> **Q: Ways to obtain prompts and number of videos used for quantitative evaluation.**
>
> A: Since no public benchmarks are available for compositional 3D-aware video generation, we first obtain sub-prompts for each expert model following their schemes. These sub-prompts are then used as key terms and composed into a complete input prompt using LLMs. For quantitative evaluation, we **randomly generated 400 videos** with varied scenes, objects, or motions, avoiding cherry-picking, and reported the average value of CLIP Score and Q-Align Score.
>
> **Q: Quantitative ablation studies should be provided.**
>
> A: We provide the ablation experiments quantitatively as follows:
>
> **Table. 1**: Quantitative comparisons of ablation studies on trajectory estimation with multi-modal LLM
> | Methods                                 | Direct Estimation. | Estimation using bounding box. | Step-by-step estimation. | Ours.  |
> |-----------------------------------------|--------------------|-------------------------------|--------------------------|-------|
> | **QAlign-img-quality** $\uparrow$       | 2.056              | 2.894                         | 3.752                    | **4.030** |
> | **QAlign-img-aesthetic** $\uparrow$     | 1.568              | 2.156                         | 3.047                    | **3.471** |
> | **QAlign-vid-quality** $\uparrow$       | 2.178              | 3.043                         | 3.904                    | **4.112** |
> | **QAlign-vid-aesthetic** $\uparrow$     | 1.680              | 2.346                         | 3.342                    | **3.723** |
> | **CLIP Score** $\uparrow$               | 25.68              | 29.84                         | 36.73                    | **38.36** |
>
>
> **Table. 2**: Quantitative comparisons of ablation studies on composition with 2D diffusion models
> | Methods                           | Without SDS. | With scale refinement. | With trajectory refinement. | Ours  |
> |-----------------------------------|--------------|------------------------|-----------------------------|-------|
> | **QAlign-img-quality** $\uparrow$ | 3.045        | 3.674                  | 3.826                       | **4.030** |
> | **QAlign-img-aesthetic** $\uparrow$ | 2.752        | 3.046                  | 3.341                       | **3.471** |
> | **QAlign-vid-quality** $\uparrow$ | 3.129        | 3.794                  | 3.983                       | **4.112** |
> | **QAlign-vid-aesthetic** $\uparrow$ | 2.704        | 3.468                  | 3.603                       | **3.723** |
> | **CLIP Score** $\uparrow$         | 31.35        | 35.27                  | 37.04                       | **38.36** |
>
> As shown in the tables above, using a direct prompt with a multi-modal LLM for trajectory estimation results in clearly unsatisfactory outcomes. Relying solely on bounding boxes to indicate object locations within the scene yields improved but still limited performance. While the step-by-step estimation strategy offers noticeable improvements, the best results are achieved by combining both approaches. Similarly, for SDS-based refinement, applying SDS incrementally to adjust scale, location, and rotation results in substantial performance improvements.
>
> **Q: The exact prompting strategy for the LLM.**
>
> A: As demonstrated in Line 136, for an input prompt, we query the LLM with the instruction: *"Please decompose this prompt into several sub-prompts, each describing the scene, objects in the scene, and the objects' motion."* From this, we obtain the corresponding sub-prompts. The creation of the scale/trajectory estimation prompt is shown in Fig. 2 in the paper.
>
> **Q: Method is limited by expert models.**
>
> A: As recognized by Reviewer 6K4s, we can easily improve the performance of our method by leveraging more powerful expert models, without the need for significant engineering work. To add other dynamic objects such as animals, we can use methods such as [1], which is a 3D Gaussian-based method that can drive animals. We will explore this in future works.
>
> [1] GART: Gaussian Articulated Template Models, CVPR 2024

---

> > ### Comment · Reviewer_7svJ · 2024-08-12
> > **Response to rebuttal**
> >
> > I thank the authors for their detailed responses. In particular the additional qualitative results are appreciated, as well as the ablations. While I still have concerns about quality (particularly in terms of prompt adherence), the rebuttal largely addresses my concerns.

---

> > > ### Author Response · Authors · 2024-08-13
> > > **Thanks for Your Response**
> > >
> > > Dear Reviewer 7svJ,
> > >
> > > Thanks for your reply! We are pleased to hear that our response has largely addressed your concerns. We would be very grateful if you could raise your rating accordingly.
> > >
> > > If you have any further questions, please feel free to reach out—we are open to continued discussion.
> > >
> > > Best,
> > >
> > > Authors

---

> ### Author Response · Authors · 2024-08-12
> **We hope that our response addresses your concern**
>
> Dear Reviewer 7svJ,
>
> We greatly appreciate the time you've invested in reviewing our paper. Having submitted our rebuttal, we are eager to know if our response has addressed your concern. As the end of the rebuttal phase is approaching, we look forward to hearing from you for any further clarification that you might require.
>
> Best,
>
> Authors

---

### Official Review · Reviewer_N95p · 2024-07-14

**Soundness:** 2
**Presentation:** 3
**Contribution:** 2
**Rating:** 5
**Confidence:** 5

**Summary:**

In this work, the authors propose a framework for 3D-aware video generation using guidance from LLM. Specifically, this work follows previous studies on LLM for video generation that takes language model as a director to do below sub-tasks:
1) expand and decompose the prompt into different aspects, and then use off-the-shelf expert models to generate objects/motions/scenes.
2) plan the trajectory and other scene configurations.
3) put all of them together and update with SDS loss.
Experiments are conducted to verify the effectiveness of this work.

**Strengths:**

1) The focused setting is novel, which enables LLM-guided text-to-video generation with 3D awareness.
2) The proposed pipeline generally makes sense to me.

**Weaknesses:**

1) The novelty of this work is limited. It seems like to be the combination of 3D scene generation + 3D avatar generation + motion generation + LLM + SDS. It would be subjected to the performance upper bound set by each expert model, and combining them altogether would make the quality even worse.
2) In TC4D, the trajectory can also be generated by LLMs, which should not be a drawback of that work.

**Questions:**

How do the authors handle the situation where scene configurations are not properly

**Limitations:**

The limitations are discussed.

---

> ### Author Rebuttal · Authors · 2024-08-06
>
> Dear Reviewer N95p, thank you for taking the time to review our work and for providing thoughtful feedback. We are pleased that you found our setting novel and the pipeline coherent. We address your concerns as follows.
>
> **Q: Novelty of this work.**
>
> A: As noted by Reviewers 7svJ and BLtC, our work transcends a mere combination of expert models. It is rooted in the concept that our understanding of the world is inherently compositional, and we achieve this by leveraging priors from expert models. Specifically, to seamlessly combine different models into a cohesive whole, we introduce a novel method that employs LLMs as a step-by-step director, capable of generating plausible trajectories with only the background scene. Subsequently, to enhance the details of the composed dynamic scenes, we propose using 2D diffusion priors (i.e., SDS) to ensure that the rendered images more closely match natural image distributions. Additionally, as acknowledged by Reviewer 6K4s, our pipeline can be easily enhanced by developing better modules for each component, highlighting the potential of our method.
>
> **Q: Trajectory in TC4D.**
>
> A: Thank you for bringing this to our attention, and we apologize for any confusion caused. As an outstanding approach for text-to-4D generation, TC4D enables trajectory-conditioned 4D creation, accommodating trajectories that are either user-defined or estimated directly by LLMs, and facilitating applications such as generating 4D scenes from arbitrary trajectories and synthesizing compositional 4D scenes. Our method differs from TC4D in several key aspects: **1)** we use 3D Gaussians instead of a deformable NeRF for representing 4D scenes, as 3D Gaussians offer real-time rendering and easier editing; **2)** for trajectory estimation, we query the LLM in an step-by-step manner, which can estimate reasonable trajectories using only the background scene; **3)** in addition to object-level composition, our method achieves composition at the scene level, including interactions between complex scenes and objects, marking a novel advancement in scene-level composition. We will make the necessary corrections in a future version to address this issue and ensure that our work meets the highest standards of accuracy and clarity.
>
> **Q: Solutions when scene configurations are not properly.**
>
> A: Based on the estimated scene configurations provided by LLMs, we will use SDS for refinement to ensure the composed 4D scene renders more realistic images that align with human intuition. Specifically, in this paper, we treat properties such as scale, location, and rotation as optimizable variables, refined by SDS. In the future, we plan to consider additional factors, such as illumination and spherical harmonic coefficients, to achieve more realistic video generation.

---

> > ### Comment · Reviewer_N95p · 2024-08-12
> >
> > I thank the authors for the reply and I acknowledge that I've read the rebuttal. Partial of my concerns are resolved, but the novelty and contribution is still kind of limited to me after reading other reviewer colleagues' comments. I hence maintain my score.

---

> > > ### Author Response · Authors · 2024-08-13
> > > **Thanks for Your Response**
> > >
> > > Dear Reviewer N95p,
> > >
> > > Thanks for your reply! We truly value your involvement in this discussion. We are happy to know that our response has resolved your concerns.
> > >
> > > If you have any further questions, please feel free to reach out—we are open to continued discussion.
> > >
> > > Best,
> > >
> > > Authors

---

### Official Review · Reviewer_6K4s · 2024-07-15

**Soundness:** 3
**Presentation:** 3
**Contribution:** 3
**Rating:** 5
**Confidence:** 3

**Summary:**

The paper presents a pipeline for 3D-aware video generation by composing scenes, objects, and motions. One key idea is to use existing LLM to provide coarse guidance on the scale and trajectory of objects, and then refine the coarse rendering with SDS. The method is compared with multiple methods.

**Strengths:**

The idea is explainable as it composes the scene, objects, and motions in an explicit manner. So potentially it is easy to improve the pipeline by building better modules for each component.

The paper shows both quantitative results and qualitative examples. The paper is overall easy to read.

**Weaknesses:**

$\textbf{Method}$

1. It would be great to clarify what are considered to be the main contributions of the paper. The pipeline uses multiple existing modules for 3D generation, and then using LLM to generate a series of bounding boxes for the guidance of the rendering. The idea of using LLM to generate a trajectory does not look novel enough given the existing volume of literature.

2. For the refinement stage, is the scale and 3D location refined for each image individually? If so, how does the scale and location maintain temporal consistency?

3. How does the motion be compatible to the scene? Suppose it's a rough terrain or mountain area, the motion has to be adjusted?

4. To generate the bounding box trajectory, is it correct that a single image is used as input, which means no 3D information is used for LLM? If the 3D scene is existing, do you think adding 3D information and generating 3D locations could be a better option rather than 2D bounding boxes?


$\textbf{Experiments}$

1. One concern is that the overall quality of the human actors is still limited. Since the resolution is not high enough, it is hard to tell whether the method is harmonizing the image well given the examples presented.

2. It is also hard to see the 3D viewpoints changes given the examples, it seems like for most examples, the viewpoints are not changing much for the sequence of renderings. I imagine identity-preserving character inpainting methods can show similar examples? Maybe show more examples demonstrating the uniqueness of the proposed method.

3. The baselines do not include the strong motion prior as the proposed method so I am not sure if this is fair. How does the method compare to methods with explicit 3D priors [1]?

[1] Image Sculpting: Precise Object Editing with 3D Geometry Control. CVPR 2024.

**Questions:**

I think the paper tackles an interesting problem but the effectiveness of the solution remains to be justified.

---

> ### Author Rebuttal · Authors · 2024-08-06
>
> Dear Reviewer 6K4s, we are grateful for your careful review and the valuable feedback that you provided for our paper. We appreciate that you found our paper easy to read and our ideas explainable. We hope the following comments address your concerns.
>
> **Q: Main contributions of this paper.**
>
> A: As recognized by Reviewer 7svJ and Reviewer BLtC, the main contributions of this paper can be summarized as follows:
>
> 1. We approach video generation as a compositional task by leveraging existing strong domain-specific models. To the best of our knowledge, this is the first work that can realize video/4D generation from the perspective of compositional scene, object, and motion generation.
>
> 2. To compose different models into a harmonic whole automatically, we provide a novel method that utilizes LLMs as a director in a step-by-step manner, which is able to generate reasonable trajectories based solely on the background scene.
>
> 3. To further refine details of composed dynamic scenes, we propose leveraging 2D diffusion priors (i.e., SDS) to ensure the rendered images align more closely with natural image distributions.
>
> 4. We have conducted extensive experiments to demonstrate the effectiveness of our method, showing that high-fidelity videos with diverse motion and flexible control over each concept can be achieved from text.
>
> **Q: Is the scale and 3D location refined for each image individually?**
>
> A: No. Since the scale describes the relative size of the object within the 3D scene, it should remain consistent across different time steps. Therefore, we refine the scale for the first frame and apply it to all subsequent frames. For the 3D location, we optimize it for each image individually to achieve better composition.
>
> **Q: How to accommodate both scenes and motions?**
>
> A: Since the motion is generated using expert models, complex object motions will be produced for more intricate scenes. We then use SDS to enhance the compatibility of these motions with the scenes.
>
> **Q: Whether 3D information is used for LLM?**
>
> A: No. In this paper, we use only a single image as input without 3D information available. However, we believe that injecting 3D information into LLM would be beneficial for better composition, which we will explore in the future. Thank you for pointing this out!
>
> **Q: The resolution is limited.**
>
> A: In this paper, all generated videos have a resolution of 512 × 512, which is higher than that of many previous methods (e.g., Comp4D, 4D-fy, etc.). We also provide more examples in the attached PDF file to demonstrate the effectiveness of our method.
>
> **Q: Hard to see 3D viewpoints changing.**
>
> A: In this paper, we focused on highlighting the composition result by cutting out the region of the moving object, which may have made viewpoint changes less noticeable. We have included additional examples with more varied viewpoints in the attached PDF. Furthermore, we believe that applying identity-preserving character inpainting methods directly may not be effective, as 3D consistency is difficult to ensure.
>
> **Q: Comparisons with methods using different motion priors.**
>
> A: Actually, methods like Comp4D and 4D-fy also use strong implicit motion priors derived from pre-trained video diffusion models, whereas we use explicit motion priors to achieve better composition and controllability. In Image Sculpting, 2D objects are projected into 3D for image editing, we share a similar concept on 3D structural representation, but focuses on composing various 3D concepts into a 4D scene for video generation. Since Image Sculpting is not designed for video generation, a direct comparison is challenging. We plan to explore integrating methods from Image Sculpting in the future to incorporate more robust motion priors.

---

### Author Rebuttal · Authors · 2024-08-06

We extend our sincere thanks to all the reviewers for their time and effort. We appreciate your positive feedback, noting that our work was described as "novel and elegant" (N95p, 7svJ, BLtC), "effective and achieving better results" (N95p, 7svJ, BLtC), and "easy to read and well-organized" (6K4s, 7svJ). In response to your comments, we have addressed each reviewer's concerns individually and uploaded a PDF file with additional qualitative examples showcasing the superiority of our proposed method. We would be grateful if you would consider increasing your score based on our responses.

---

### Decision · Program_Chairs · 2024-09-25

**Decision:**

Accept (poster)

**Comment:**

This paper proposes a method for compositional 3D-aware video generation from text prompt. The main contribution is a meticulous design and leverage of existing LLM and diffusion models for object, motion, scene synthesis. Reviewers are positive about the rationality of the overall framework and competitive performance compared to baselines. During rebuttal, reviewers raised concerns about the novelty, insufficient evaluation and questions about other technical details. Most of the concerns have been addressed by the rebuttal and reviewers lean to the positive side of the paper, thus this paper is recommended to be accepted. We encourage the authors to include all additional results in the revision. Moreover, reviewers have raised reasonable concerns about the limitation (e.g., interaction between scene and character) of the current method and these shall also be discussed in the paper.